# Synthesis of Chemically Cross-Linked pH-Sensitive Hydrogels for the Sustained Delivery of Ezetimibe

**DOI:** 10.3390/gels8050281

**Published:** 2022-05-01

**Authors:** Rahima Khan, Muhammad Zaman, Ahmad Salawi, Mahtab Ahmad Khan, Muhammad Omer Iqbal, Romana Riaz, Muhammad Masood Ahmed, Muhammad Hammad Butt, Muhammad Nadeem Alvi, Yosif Almoshari, Meshal Alshamrani

**Affiliations:** 1Faculty of Pharmacy, University of Central Punjab, Lahore 54000, Pakistan; rahima.khan888@gmail.com (R.K.); mahtab.ahmad@ucp.edu.pk (M.A.K.); nadeem.alvi@ucp.edu.pk (M.N.A.); 2Department of Pharmaceutics, College of Pharmacy, Jazan University, Jazan 45142, Saudi Arabia; asalawi@jazanu.edu.sa (A.S.); yalmoshari@jazanu.edu.sa (Y.A.); malshamrani@jazanu.edu.sa (M.A.); 3Shandong Provincial Key Laboratory of Glycoscience and Glycoengineering, School of Medicine and Pharmacy, Ocean University of China, Qingdao 266005, China; oiqbal133@gmail.com; 4Department of Pharmaceutics, Faculty of Pharmacy, Bahauddin Zakariya University Multan, Multan 59300, Pakistan; rummanariaz@yahoo.com; 5Faculty of Pharmaceutical Sciences, Times Institute, Multan 59300, Pakistan; masoodkarni@yahoo.com

**Keywords:** Ezetimibe, synthesis, crosslinking, pH sensitive, hydrogels, sustained release

## Abstract

In recent years, pH-sensitive hydrogels have been developed for the delivery of therapeutic agents to specific target sites that have a defined pH range. The use of pH-responsive polymers in hydrogels allows drug delivery to the desired pH range of the target organ. The primary aim is to increase the retention time of the drug in the small intestine by utilizing the swelling mechanism of the hydrogel at intestinal pH. In this study, polyethylene glycol (PEG) was used as a polymer to formulate a pH-sensitive hydrogel of Ezetimibe to deliver the drug to the small intestine where it inhibits the absorption of cholesterol. Design Expert software was applied to design and optimize the trial formulations in order to obtain an optimized formulation that has all the desired characteristics of the hydrogels. The PEG/Acrylic Acid hydrogels showed the maximum swelling at pH 6.8, which is consistent with the pH of the small intestine (pH 6–7.4). The maximum entrapment efficiency of the hydrogels was 99%. The hydrogel released 80–90% of the drug within 24 h and followed first-order release kinetics, which showed that the release from the drug was sustained. Hence, the results showed that the choice of a suitable polymer can lead to the development of an efficient drug-loaded hydrogel that can deliver the drug at the specific pH of the target organ.

## 1. Introduction

Recent developments in modern pharmaceutical science have resulted in the use of hydrogels as effective drug-delivery systems for both hydrophilic and hydrophobic drugs [1]. Hydrogels, having a three-dimensional crosslinked network, are a type of novel drug-delivery system that is capable of absorbing and holding large quantities of water [2]. Hydrogels are like soft tissues, and their functionality also resembles that of soft tissues. They are compatible with blood components. 

These materials are not only important for the field of medical sciences but also for applications used in the biomedical domain [3]. Macromolecules, such as polymers, have been used for hydrogels that are sensitive to the pH level, and these have received more attention as newly formulated material used for intestinal targeting. This is due to their non-toxic and biocompatible nature [4].

Recently, polyethylene glycol (PEG)-based hydrogels have been used extensively and have been investigated for many biomedical applications. These include various functionalities, such as the delivery of drugs, cellular matrix and coating of the base, because PEG is soluble in water, biologically compatible and non-toxic for the cellular structure and matrix. Normally, PEG has hydroxyl groups attached at the significant end of the chain, which is more compatible with other functional groups [5]. Different procedures have been used to fabricate the chemical combinations for the hydrogels based on PEG. The hydrogel network is easily handled and controlled due to the length, type and ligand-binding capabilities, and this provides significant benefits of PEG-based hydrogels that are needed for the manipulation of drug carriers [6].

Ezetimibe is a drug belonging to the class of lipid agents that stops the absorption of cholesterol found in diet and that present in biliary secretions without disturbing the absorption of the vitamins that are fat-soluble and other elements, such as triglycerides (TG) [7]. It is applied in the treatment of familial hypercholesterolemia and even in the case of non-familial hypercholesterolemia as well as in the area of adjunctive monotherapy of homozygous familial sitosterolemia [8,9].

The curative impact of the drug, i.e., Ezetimibe, is related to the period in which it resides in the intestinal tract. The longer it resides in the intestinal tract, the greater its ability to stop the absorption of cholesterol. To achieve this objective, a sustained-release pH-sensitive hydrogel of Ezetimibe was formulated that is capable of releasing the drug slowly, thus, providing the drug with a better opportunity to inhibit the absorption of cholesterol from the intestine and, hence, increasing therapeutic efficiency. This statement has led to the conclusion that oral delivery of Ezetimibe in the form of a hydrogel will surely improve its therapeutic effects by increasing its retention time in the intestines.

## 2. Results and Discussion

### 2.1. Physical Appearance

The chemically-crosslinked hydrogels were prepared using the free radical polymerization technique. The unloaded PEG/Acrylic Acid and β-CD/Acrylic Acid hydrogels were transparent in color. When immersed in the drug solution of Ezetimibe, the gels became completely white in color. This is because Ezetimibe is white in color. Reddy, Nagabhushanam et al. created Ezetimibe-loaded hydrogel beads that appeared to be white in color after drug loading [10]. The gels were completely homogeneous in shape, color and surface. There were no signs of discoloration, aggregation or unwanted spots on the surface of the blank hydrogel (A) and drug-loaded hydrogel (B) (Figure 1). 

### 2.2. Numerical Optimization

The optimized formulations of PEG/Acrylic Acid hydrogels and β-CD/Acrylic Acid hydrogels were studied for all evaluation parameters. The results of the evaluation parameters of optimized formulations of PEG/Acrylic Acid hydrogels were compared with the predicted outcomes of these evaluation tests obtained from Design Expert software. The results showed no significant difference from predicted outcomes and the optimized formulation was considered to be a successful formulation of PEG/Acrylic Acid hydrogels having all the required characteristics of optimal porosity, gel fraction, degree of swelling, entrapment efficiency and in vitro drug release. A comparison of the predicted results and the results obtained from the evaluation parameters in the optimized formulation are shown in Table 1.

### 2.3. Swelling Studies

Swelling studies of PEG/acrylic acid hydrogels were performed in pH 1.2, 6.8 and 7.2. The PEG/Acrylic Acid hydrogels showed the maximum swelling in pH 6.8 buffer solution. The hydrogels showed a minimal degree of swelling in a buffer solution at pH 1.2 (Figure 2).

The trial formulation of the hydrogels contained varying quantities of polymer (PEG), monomer (acrylic acid) and crosslinker (MBA). The concentration of potassium per sulfate (KPS) was the same in all formulations. The effects of changing the concentrations of polymer, monomer and crosslinker on the degree of swelling of the hydrogels were studied. All the hydrogels showed pH-dependent swelling. Significant differences in the swelling of the hydrogels at acidic and basic pH were observed.

PEG/Acrylic Acid hydrogels showed a prominent increase in the degree of swelling as the pH increased. The gels showed the maximum swelling at pH 7.2. This trend can be attributed to the fact that, as the pH increases, the degree of ionization of carboxyl groups increases; hence, there is greater interaction between PEG and acrylic acid resulting in greater swelling. Ijaz et al. created PEG/acidic acid hydrogels that showed the maximum swelling at pH 6.8 indicating similar results [11]. Studies showed that, at a lower pH, the repulsion between anionic groups of PEG and acrylic acid reduces, which results in decreased swelling. This is due to the fact that, at low pH, most carboxylic groups in the hydrogel are protonated [12,13] (Figure 3).

### 2.4. Sol-Gel Fraction

Sol–gel fraction analysis of the hydrogels was conducted to determine the fraction of uncrosslinked polymer in the hydrogel. The PEG/Acrylic Acid hydrogels showed a sol fraction and gel fraction in the range of 0.3–12.2% and 87.7–99.6%, respectively. The gel fraction of hydrogels was determined to investigate the amount of uncrosslinked polymer left in the hydrogel. An increase in the gel fraction was observed by increasing the concentration of PEG, AA and MBA. 

This is due to the fact that more polymers and monomers provide a greater number of chains for crosslinking, and more crosslinkers available increases the proportion of crosslinking; hence, the amount of uncrosslinked reactants is less. These findings are in agreement with those reported by Sarfraz et al., where they developed a polymeric nano hydrogel of naproxen sodium and found an increase in gel fraction with an increase in concentration of polymer, monomer and crosslinker [14] (Figure 4).

### 2.5. Porosity

The porosity of the hydrogels was checked to verify the degree of porous structure of the gels. The PEG/Acrylic Acid hydrogels showed a percentage porosity in the range of 101.54–243.19% (Figure 5). The porosity of the hydrogel formulations was determined to check the fraction of volume of pores in the hydrogel. The results of the porosity measurement showed that an increase in the concentration of polymer (PEG) and monomer (acrylic acid) resulted in an increase in the porosity of the hydrogels. Bukhari et al. developed chemically crosslinked hydrogels of gelatin and acrylic acid and reported similar results. 

An increase in the porosity of hydrogels due to an increase in the concentration of monomer and polymer is due to an increase in the viscosity of the hydrogel solution. Greater viscosity results in more interconnected channels, which, in turn, increases the porosity of the solution [15]. An increase in concentration of crosslinker resulted in decreased porosity due to entanglement of monomer and polymer chains resulting in a rigid hydrogel. Similar results were reported by Yin et al., who formulated super porous hydrogels of acrylic acid and acrylamide [16]. 

### 2.6. Drug-Entrapment Efficiency

The amount of drug entrapped in 2 mm disks of the hydrogel was calculated in terms of a percentage. The drug-entrapment efficiency of PEG/Acrylic Acid hydrogels was in the range of 38.04–99.84%. The physical and chemical properties of the drug, as well as the hydrogel, have a significant impact on the entrapment efficiency of the hydrogel [17]. The influence of concentrations of PEG, AA and MBA on the entrapment efficiency of the hydrogels was observed. The entrapment efficiency of the hydrogels increased with an increase in the concentration of the polymer and monomer. 

This is because a greater amount of polymer and monomer create a wider hydrogel network and thus provide more entrapment opportunities for the drug [18]. Hence, greater amounts of polymer and monomer lead to greater entrapment, which is in agreement of the findings of Mahmood et al. [19]. In contrast to the polymer and monomer, increases in the concentration of crosslinker decreased the entrapment efficiency of the hydrogel. This is because a greater amount of crosslinker renders a rigid hydrogel with reduced porosity and swelling capability. This limits the permeation of the drug through the hydrogel surface, and hence the entrapment efficiency decreases. Malik et al. reported similar results in their study in which they formulated hydrogels based on chitosan and xanthan gum for the delivery of acyclovir [20] (Figure 6).

### 2.7. In Vitro Drug Release Studies

In vitro drug release studies were conducted in a USP Dissolution apparatus by using a phosphate buffer of pH 7.2 as dissolution medium. The in vitro drug release behavior of the hydrogel formulations was observed for 24 h in a phosphate buffer of pH 7.2. All formulations showed 90–100% release within 24 h. As seen in a study conducted by Ramadan et al., drug release from the hydrogels was found to be dependent on the concentrations of polymer, monomer and crosslinker [21] (Figure 7).

### 2.8. Drug-Release Kinetics

To determine the release pattern of Ezetimibe from the hydrogel disks, the results of in vitro drug release studies were subjected to DD Solver software to assess the R2 values for the Zero Order, First Order, Higuchi model, Korsmeyer–Peppas model and Hixson–Crowell mode. The results showed that the R^2^ value of the Hixson–Crowell model was highest for the PEG/Acrylic Acid hydrogels. The n value of the hydrogel formulations was greater than 0.5, which shows that the pattern of drug release follows non-Fickian diffusion (Table 2).

### 2.9. Fourier-Transform Infrared Spectroscopy

Fourier-transform infrared analysis was conducted in order to analyze the presence of functional groups in the hydrogel components and to determine the functional groups responsible for achieving the crosslinking mechanism. FTIR spectra of the pure drug [7], PEG 6000, acrylic acid, KPS, MBA, drug-loaded and unloaded hydrogel were conducted. The FTIR spectrum of Ezetimibe showed peaks at 3265, 1725, 1507, 1212 and 1063 cm^−1^ that represented -OH, C = O, C = C, C-F and C-O stretching vibrations. 

All these peaks were also visible in the drug-loaded hydrogel formulations, which showed that the drug was compatible with all other excipient and that there was no sort of incompatibility in the constituents of PEG/Acrylic Acid hydrogels [22]. The FTIR spectrum of PEG 6000 shows a strong peak at 2887 cm^−1^, which corresponds to C-H stretching vibrations. The peak at 1464 cm^−1^ shows C-H bending vibrations, while the peak at 1297 cm^−1^ shows O-H stretching vibrations. The C-O-C and C-O-H stretching vibrations are depicted by the peaks at 1109 cm^−1^ and 1094 cm^−1^, respectively [23]. 

The spectrum of Acrylic Acid shows an -OH bond peak at 2987 cm^−1^ and -COOH stretching at 1760 cm^−1^. The peak at 1694 cm^−1^ indicates C = O stretching vibrations, and the one at 1634 cm^−1^ depicts C = C stretching vibrations [24]. The major peak in the FTIR spectrum of Potassium Per Sulfate is the one at 1380 cm^−1^ that indicates the S = O (Sulfate bond) stretching vibrations [25]. The FTIR spectrum of MBA shows a very prominent peak at 3305 cm^−1^ that shows N-H stretching vibrations. 

The peak at 1560 cm^−1^ shows C = O stretching vibrations, and the one at 1535 cm^−1^ shows N-H deformation. C-N stretching vibrations are indicated by the peak at 1301 cm^−1^. The peak at 965 cm^−1^ corresponds to N-C’ stretching vibrations, and the one at 955 cm^−1^ shows C-C_α_ stretching vibrations. The O = C N vibrations are represented by the peak at 627 cm^−1^, and the C-C_α_ bending vibrations are shown by the peak at 293 cm^−1^ [26] (Figure 8).

### 2.10. Scanning Electron Microscopy

Scanning electron microscopy was conducted to study the surface morphology of the crosslinked hydrogel networks for drug-loaded PEG/Acrylic Acid hydrogels. The SEM images obtained showed that both types that hydrogels showed a porous structure with a course and wavy surface (Figure 9).

### 2.11. X-ray Diffraction

X-ray diffraction analysis was conducted for pure drug and drug-loaded PEG/Acrylic Acid hydrogel. The diffraction patterns of Ezetimibe and drug-loaded formulations were compared. The XRD graph of Ezetimibe showed a sharp peak at an angle of 20°, which indicated the crystalline nature of pure drug. However, the diffractograms of the drug-loaded hydrogels showed diffused peaks instead of sharp peaks, which indicated that the drug was present in an amorphous morphology in the formulated hydrogel that appeared to be in the form of a solid solution. The diffused nature of the peaks for drug-loaded hydrogels confirmed the entrapment of the drug in the chemically crosslinked hydrogel network and the masking of the crystalline form of Ezetimibe into an amorphous-like form (Figure 10).

## 3. Conclusions

The goal of the study was the formulation of a pH-sensitive hydrogel of Ezetimibe using a pH sensitive polymer that swells at the intestinal pH and increases the retention time of the drug. Polyethylene glycol acted as an efficient pH-sensitive polymer and allowed the hydrogel to swell at pH 6.8 and 7.2 at a maximum rate, which means that the hydrogel will swell and retain the drug at the intestinal pH, which ranges from pH 6 to 7.4. Within 24 h, almost 99% of the drug was released from the hydrogels, and the release kinetics showed that the drug followed first-order kinetics and was released from the hydrogel discs in a sustained manner. The current findings were found to justify the objectives of the study—that is, to retain the drug in the intestinal tract for better therapeutic outcomes. However, future studies can put more effort in determining the efficacy of the hydrogels by in-vivo evaluation.

## 4. Materials and Methods

### 4.1. Materials

Ezetimibe was obtained as a gift sample from CCL Pharmaceuticals (Lahore, Pakistan). Polyethylene glycol 6000 (PEG), N, N’-methylene bis acrylamide (MBA), KPS and acrylic acid were purchased from Sigma Aldrich. All other excipients were analytical grade, and the distilled water used in the study was prepared in the laboratory. 

### 4.2. Methods

#### 4.2.1. Formulation Design by Design Expert

The formulations of Ezetimibe hydrogel were prepared according to a formulation deign created via Design Expert Version 11 Software. The concentration of the initiator was kept constant. The variables included the polymer (PEG 6000), the monomer (acrylic acid) and the crosslinker (N, N’-methylene bis acrylamide). The final design contained compositions of 14 trial formulations of PEG/acrylic acid hydrogels (Table 3). 

#### 4.2.2. Formulation of PEG/Acrylic Acid Hydrogels by Free-Radical Polymerization

The hydrogels were prepared by the free-radical polymerization crosslinking technique. The specified amount of the polymer (PEG 6000) was dissolved in 4–5 mL of distilled water while stirring continuously on a magnetic stirrer. The monomer was taken in a separate beaker in which the initiator (0.02 g) was added. This mixture was also stirred until homogenous. The monomer solution was then slowly added to the polymer solution with continuous stirring. After the formation of a homogenous mixture, the crosslinker was added at this point. 

The solution was stirred for some time, and finally the volume was increased to 10 mL with distilled water. The solution was then continuously stirred until everything was combined and the mixture appeared transparent and homogenous. This mixture was then transferred to a test tube and placed in a water bath at 50 °C for 3 h to allow the hydrogel to solidify. The test tube was then removed from the water bath and allowed to cool down at room temperature. 

Then, the gel was removed from the test tube and cut into 1.5 cm long pieces. These pieces were washed with a 70:30 mixture of ethanol and water. The pieces were then placed in a Petri dish and allowed to dry in a drying oven at 40 °C for 7 days. The prepared gels were then removed from the oven and subjected further to drug loading [27]. The reaction mechanism of formation of the crosslinked PEG/Acrylic Acid hydrogel is given in Figure 11.

### 4.3. Drug Loading

The post-loading method of drug loading was employed in this study. The drug was loaded after the formation of hydrogel. A total of 0.1 g of Ezetimibe was weighed and dissolved in 10 mL of methanol by stirring continuously on a magnetic stirrer. To this solution, 10 mL of pH 7.2 buffer was added dropwise. The mixture was stirred continuously to obtain a homogenous drug solution. 

This solution was then transferred to an air-tight container. A 2 mm piece of the hydrogel was cut by using a paper cutter and added to this solution and allowed to sit for 7 days so that maximum amount of the drug was absorbed by the hydrogel. The swollen gel containing the drug was then removed from the container and washed with a 70:30 mixture of ethanol and water. The gel was then placed in a petri dish and allowed to dry in a drying oven at 40 °C for one day. The drug-loaded gel was then submitted to characterization tests [20]. 

### 4.4. Numerical Optimization

The gel fraction, porosity, trapping efficiency, degree of swelling and in vitro drug release were the responses studied for all formulations of both type of hydrogels. All the responses of these parameters were added to Design Expert Software, and Analysis of Variance (ANOVA) was used to validate the experimental outcomes for all 14 trial formulations of PEG/Acrylic Acid hydrogels. Various combinations were attempted in order to determine an optimized formulation with maximum desirability for both types of hydrogels. 

The criteria for numerical optimization of PEG/acrylic acid hydrogels were set by keeping the quantities of polymer, monomer and crosslinker in range. The gel fraction was maximized. Porosity and entrapment efficiency were kept in the range, while the degree of swelling and in vitro drug release were set to a specific target. This resulted in an optimized formulation having a desirability of 0.903 with a composition containing 0.1963 g of PEG, 2.523 g of acrylic acid and 0.0365 g of MBA. 

### 4.5. Characterization

The formulated hydrogels were subjected to evaluation tests for the following parameters.

#### 4.5.1. Degree of Swelling

To determine the swelling ratio of the hydrogels in different pH environments, they were immersed in buffers of pH 1.2, 6.8 and 7.2. The method to check the pH of PEG containing hydrogels was previously reported in the literature [28]. Initially, 2 mm disks were cut from each formulation. The initial weight of the disks was noted, and they were immersed in the buffer solutions. Then, the weight of the disks was noted after 1, 2, 3, 4, 24 and 48 h to determine the swelling ratio. The degree of swelling was then determined by the following formula:(1)$$\mathrm{Degree}\;\mathrm{of}\;\mathrm{Swelling}=\left[\frac{W_{1}-W_{0}}{W_{0}}\right]\times 100$$
where *W*_1_ is the weight of the swollen hydrogel and *W*_0_ is the initial weight of the hydrogel. 

#### 4.5.2. Sol–Gel Fraction

To evaluate the number of reactants consumed during the preparation of the hydrogel, sol–gel fractions of hydrogels were performed. Sol contents are the soluble unreacted contents of the polymerization reaction [29]. For this purpose, a 5 mm disk of each hydrogel formulation was cut and weighed. Then, they were immersed in 50 mL of distilled water for 48 h. The disks were then removed and dried in an oven for 7 days [30]. After that, the weight of the disks was noted, and the sol–gel fraction was determined using the following formula:(2)$$\mathrm{Sol}\;\mathrm{fraction}\;\%=\left[\frac{W_{0}-W_{1}}{W_{0}}\right]\times 100$$
Gel fraction % = 100 − sol fraction%(3)
where *W*_0_ is the initial weight of the hydrogel disk and *W*_1_ is the weight of the disk noted after immersion and drying. 

#### 4.5.3. Porosity

The porosity of the formulated hydrogels was evaluated by solvent replacement method. For this purpose, 2 mm disks of the hydrogels were cut and weighed. The disks were then placed for 24 h in pure ethanol. After the time period, the disks were removed from ethanol and blobbed with a blotting paper to remove excess ethanol present on the surface of the disks. The disks were then weighed, and the porosity was determined by the following formula:(4)$$\mathrm{Porosity}=\left[\frac{M_{2}-M_{1}}{\rho v}\right]\times 100$$
where *M*_2_ is the weight of the soaked disks, *M*_1_ is the initial weight of the disks, *ρ* is the density of absolute ethanol and *v* is the volume of the hydrogel disks.

#### 4.5.4. Drug-Entrapment Efficiency

In order to investigate the entrapment efficiency of the hydrogels, 2 mm hydrogel disks loaded with Ezetimibe were placed in 50 mL of methanol and allowed to swell for 24 h. After that, the swollen hydrogel was crushed in a mortar and added to the same methanol in which the disk was soaked. The mixture was then allowed to homogenize for 5 min at a speed of 8000× *g* rpm. The mixture was then centrifuged at a speed of 6000× *g* rpm for 10 min. The supernatant was filtered, and its absorbance was checked in a UV-Visible Spectrophotometer. The absorbance value was then used to obtain the amount of total drug recovered with the help of the calibration curve of Ezetimibe. The % entrapment efficiency was then calculated by the following formula:(5)$$\%\;\mathrm{Entrapment}\;\mathrm{efficiency}=\frac{Amount\;of\;drug\;recovered}{Amount\;of\;drug\;added}\times 100$$

#### 4.5.5. In Vitro Drug Release Studies

In vitro drug release studies were conducted on a seven-paddle USP dissolution apparatus. The drug-loaded disks were placed in 900 mL of dissolution medium (pH7.2 phosphate buffer), and the apparatus was allowed to run at 37 °C at a speed of 100 rpm. 5 mL samples were removed from the dissolution medium after 0.5, 1, 1.5, 2, 3, 4, 5, 6, 8, 12 and 24 h. 

A total of 5 mL of the dissolution medium was immediately returned to the vessel after withdrawal of each sample to replenish the aliquot. The samples withdrawn were then passed through 0.45 µm syringe filters and their absorbance was checked in a UV-Visible spectrophotometer. The amount of drug released at each time point was determined from the equation obtained from the calibration curve. The % drug release was then calculated by the following formula:(6)$$\%\;\mathrm{Drug}\;\mathrm{Release}=\frac{Amount\;of\;drug\;present\;in\;sample}{Amount\;of\;drug\;added}\times 100$$

#### 4.5.6. Drug-Release Kinetics

To study the release kinetics of Ezetimibe from hydrogel disks, the release data were fitted to various models including the first order, zero order, Higuchi model, Korsmeyer–Peppas model and Hixson–Crowell model [31]. The release kinetics were assessed using DD Solver software, and the R^2^ values were determined for each model to determine the drug release pattern of the formulations. 

#### 4.5.7. Fourier-Transform Infrared Spectroscopy

To evaluate functional groups in monomer and polymers (AA, PEG and β-CD) and to confirm the formation of crosslinked networks from MBA hydrogels, samples were analyzed by Fourier-transform infrared spectroscopy [32]. It is also used for compatibility study of drug with the polymer and excipients. The FTIR spectra of pure PEG, pure β-CD, pure acrylic acid, pure KPS, pure MBA and drug-loaded and unloaded formulations of PEG and β-CD were recorded using an FTIR spectrophotometer (Bruker Corporation). Transit Platinum ATR technique was used, and the spectra were recorded within the wavelength range of 1000–3500 cm^1^.

#### 4.5.8. Scanning Electron Microscopy

The PEG and β-CD formulations were characterized for their surface morphology. The surface morphology of the drug-loaded and unloaded hydrogels was examined separately. The samples were prepared by soaking the hydrogel disks in pH 7.2 buffer for swelling. The swollen disks were then cut into 2 mm thick hydrogel layers. These layers were then subjected to lyophilization in Zirbus Technology GmbH VaCo 2 lyophilizer (Bad Grund, Germany). The lyophilized samples were then subjected to SEM to check the surface morphology of the hydrogels.

#### 4.5.9. X-ray Diffraction

X-ray diffraction (XRD) is a tool for determining the molecular structure and crystallinity of a material by obtaining information about the lattice parameters. The principle is to bombard the sample with an X-ray beam with different incoming angles to generate a diffraction pattern. XRD was used to determine the crystallinity of Ezetimibe-loaded PEG/Acrylic Acid hydrogels. XRD analysis can provide clues about the crystalline or amorphous nature of the drugs after loading in a hydrogel. XRD patterns were recorded for drug-loaded and unloaded hydrogels using an X-ray diffractometer [33]. 

## Figures and Tables

**Figure 1 gels-08-00281-f001:**
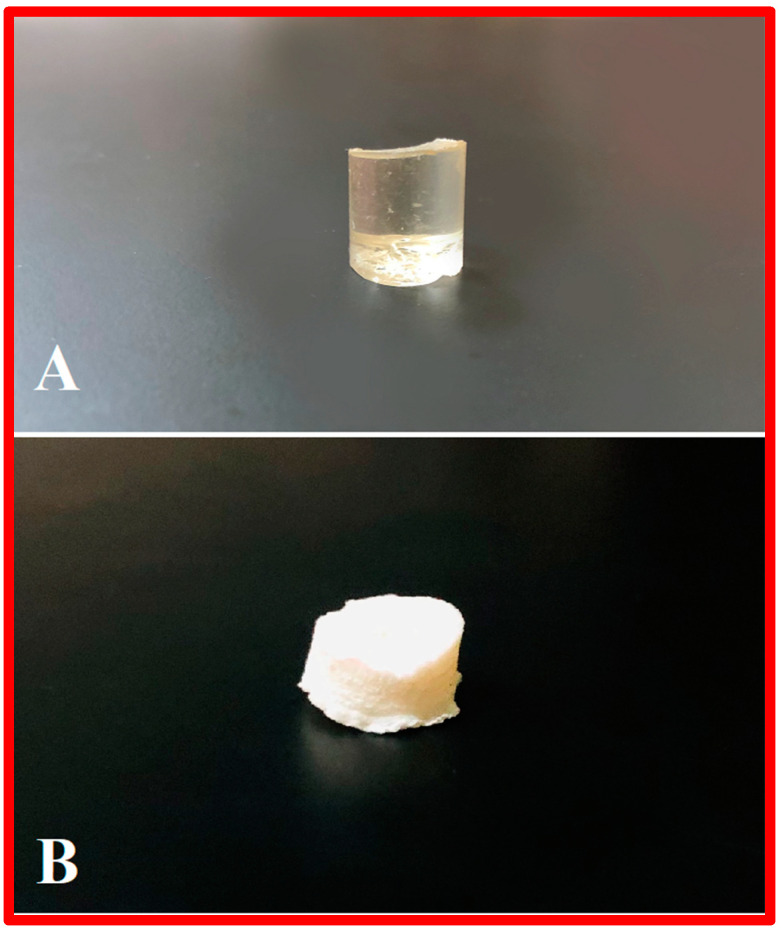
Physical appearance of a blank hydrogel (**A**) and drug-loaded hydrogel (**B**).

**Figure 2 gels-08-00281-f002:**
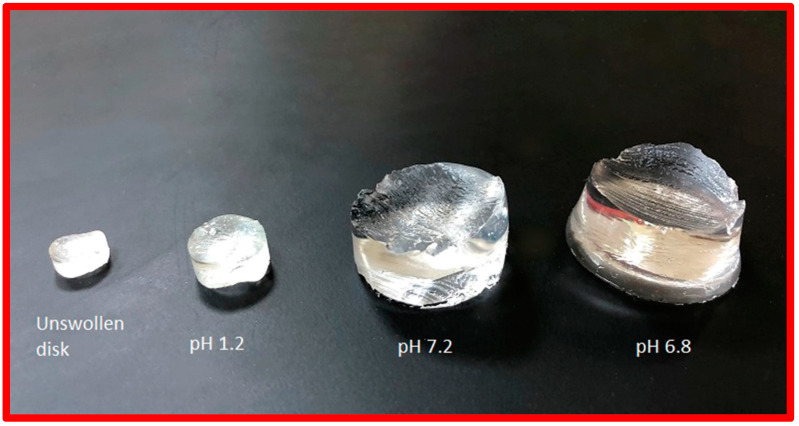
The swelling behavior of blank hydrogels at pH 1.2, 7.2 and 6.8.

**Figure 3 gels-08-00281-f003:**
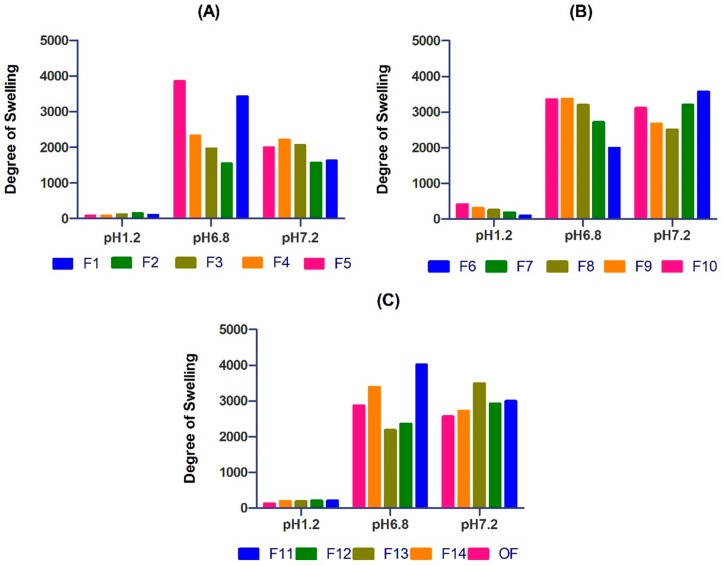
Describing the pH dependent, swelling behavior of hydrogel from F1–F5 (**A**), F6–F10 (**B**), and F11–F14, and OF (**C**) showing better swelling at higher pH.

**Figure 4 gels-08-00281-f004:**
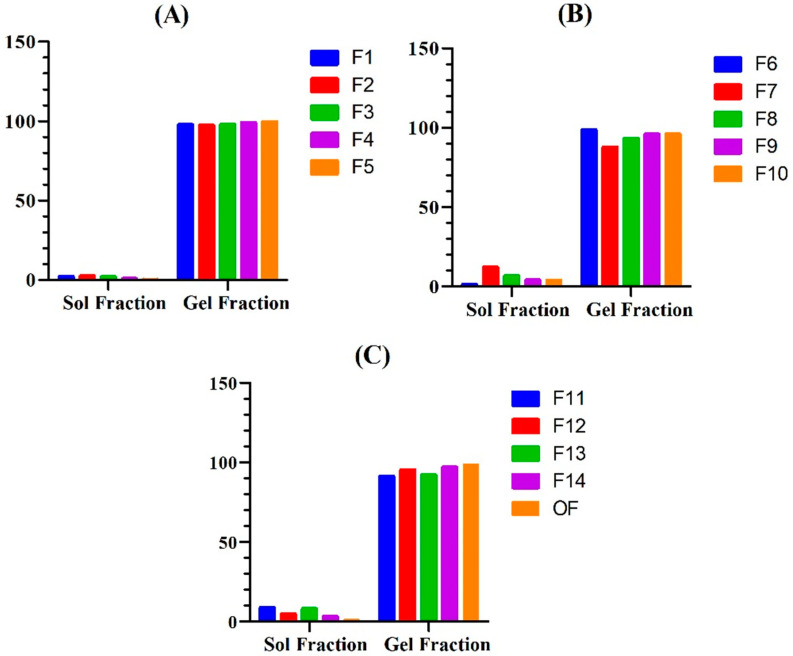
Illustrating the considerable gel fraction and minimal sol fraction of the prepared hydrogel formulation F1–F5 (**A**), F6-F10 (**B**), and F11–F14, and OF (**C**).

**Figure 5 gels-08-00281-f005:**
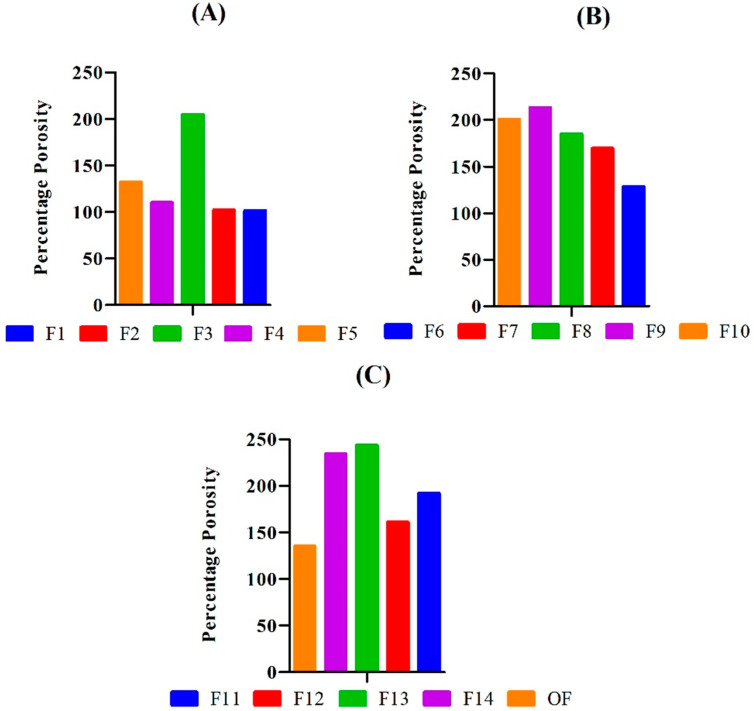
Showing greater porosity in the hydrogel formulations F1–F5 (**A**), F6–F10 (**B**), and F11–F14, and OF (**C**) having low concentration of cross-linker.

**Figure 6 gels-08-00281-f006:**
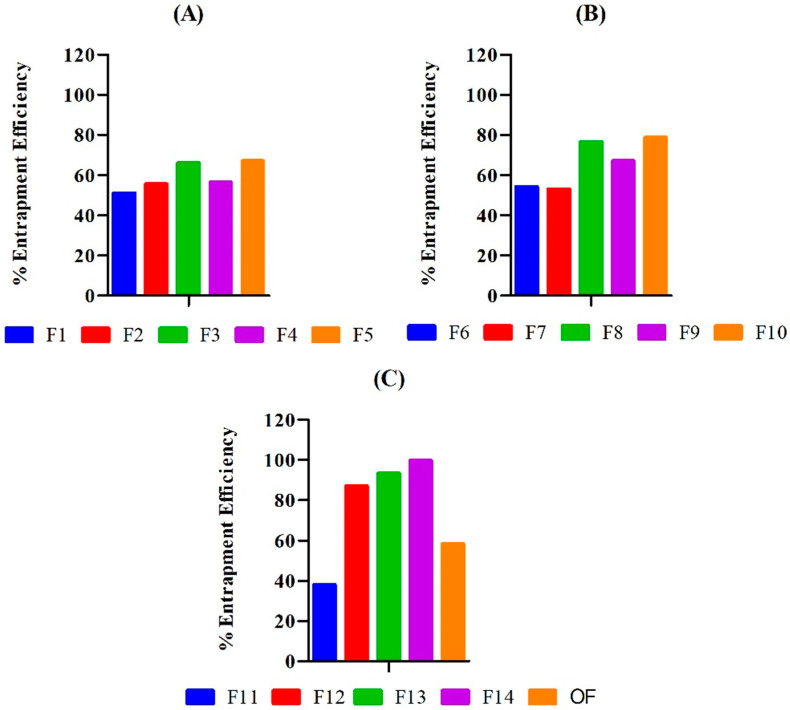
Illustrating polymer and monomer’s concentration dependent entrapment efficiency in formulation F1–F5 (**A**), F6–F10 (**B**), and F11–F14, and OF (**C**) with greater concentration of polymer and monomer have shown better entrapment efficiency.

**Figure 7 gels-08-00281-f007:**
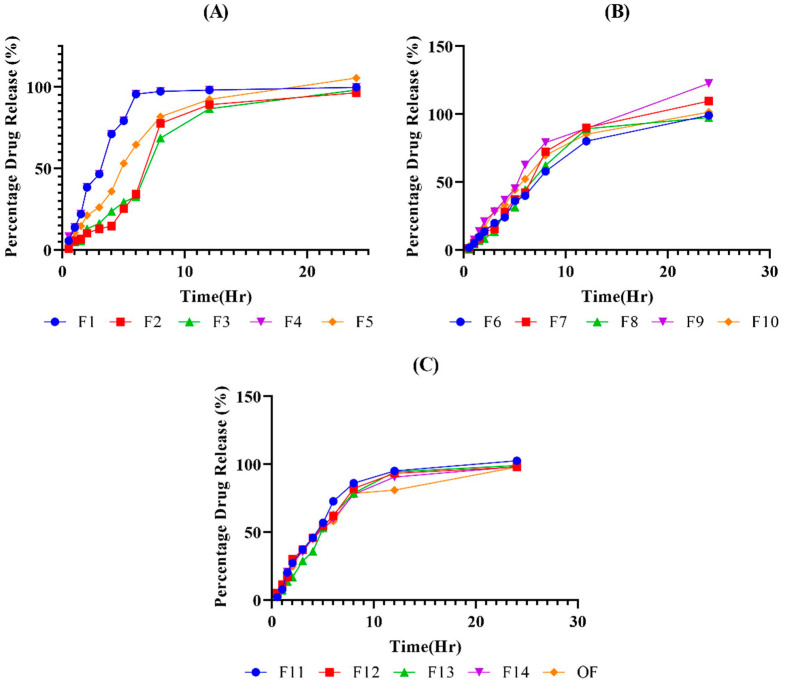
(**A**) describing in vitro drug release from F1–F5, (**B**) F6–F10, and (**C**) F11–F14, and OF. Where release pattern represents a good sustained release behavior of the drug could been observed.

**Figure 8 gels-08-00281-f008:**
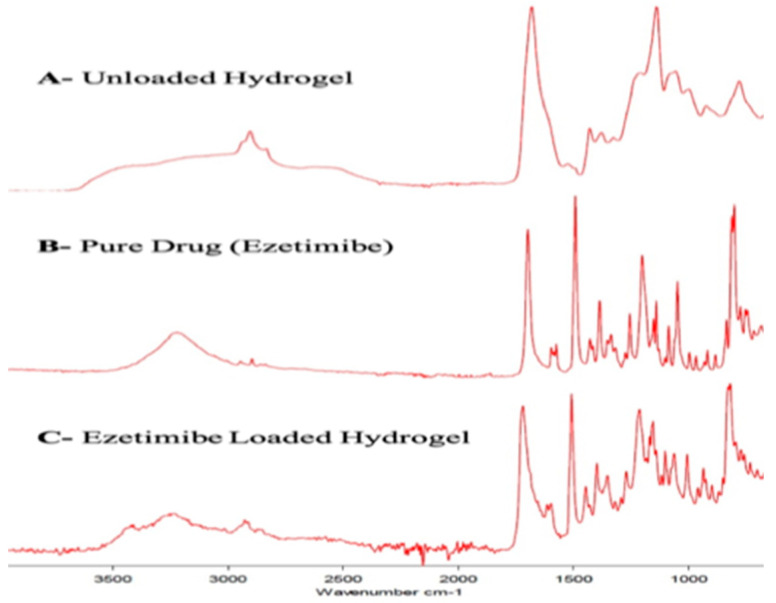
The Fourier-transform infrared analysis spectra of (**A**) unloaded hydrogel, (**B**) pure drug (Ezetimibe) and (**C**) drug-loaded hydrogel.

**Figure 9 gels-08-00281-f009:**
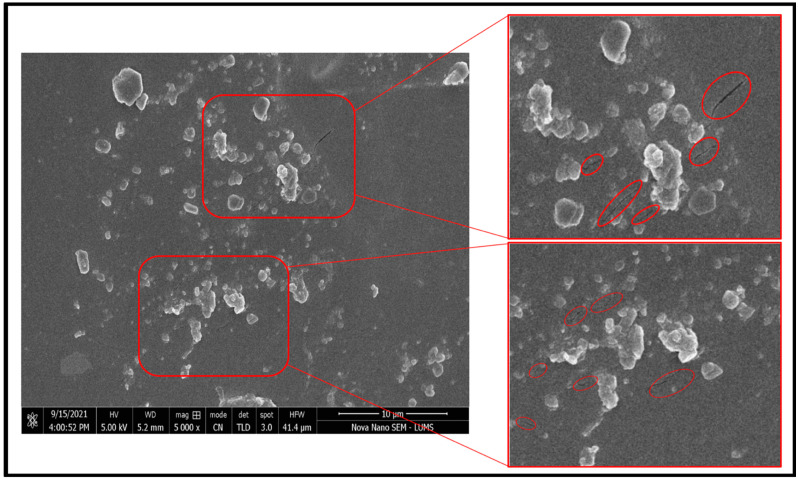
SEM images of the optimized hydrogel showed a porous structure.

**Figure 10 gels-08-00281-f010:**
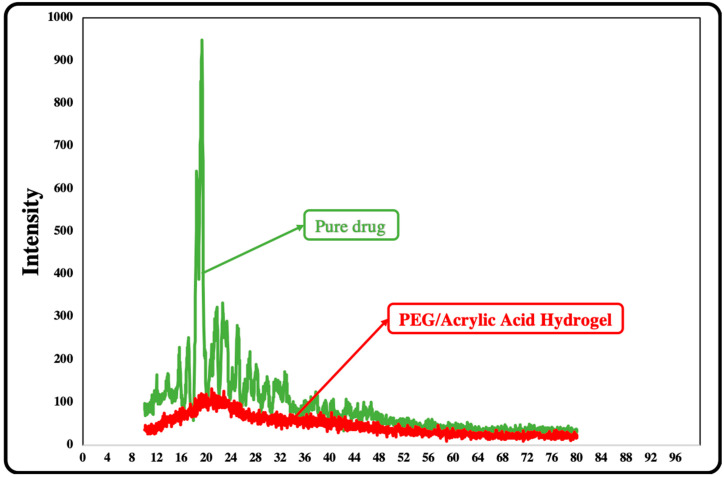
X-ray diffraction analysis of pure drug and PEF/Acrylic Acid hydrogels.

**Figure 11 gels-08-00281-f011:**
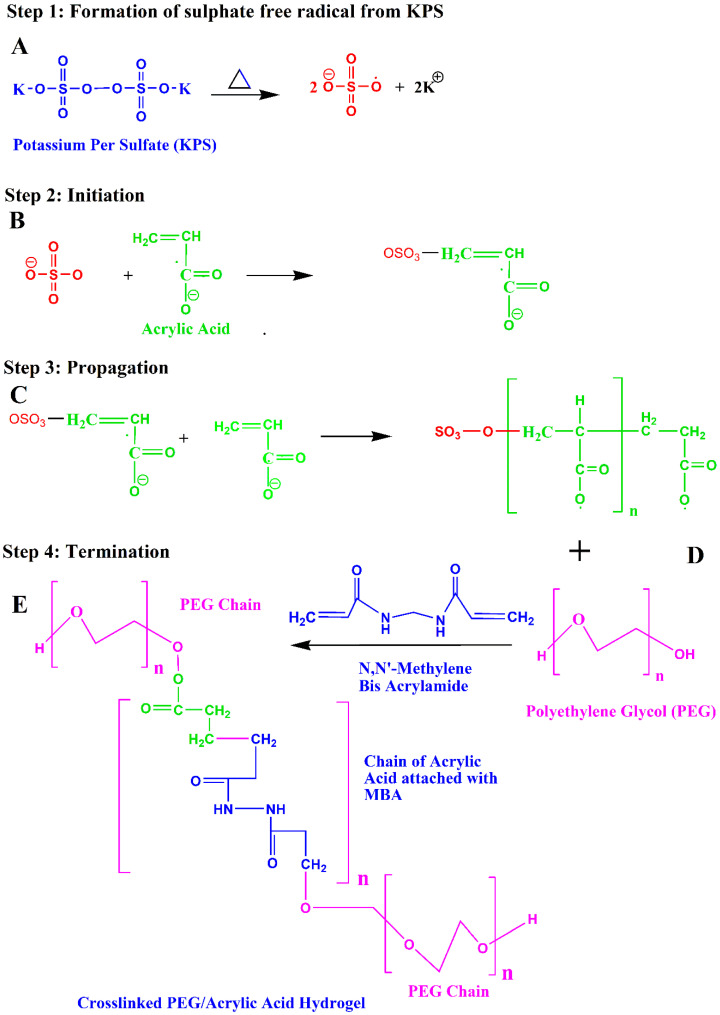
Reaction mechanism of crosslinked PEG/Acrylic Acid hydrogel.

**Table 1 gels-08-00281-t001:** Numerical optimization of the predicted outcomes and obtained results of the optimized PEG/Acrylic Acid hydrogel.

Parameter	Predicted Outcomes	Obtained Results
Porosity	134.970	135.34
Gel Fraction	97.159	98.66
Entrapment Efficiency	59.588	58.39
Degree of Swelling pH 1.2	112.020	120.93
Degree of Swelling pH 6.8	2999.997	2863.88
Degree of Swelling pH 7.2	2744.706	2556.75
In Vitro Drug Release	100 in 24 h	98.1% in 24 h

**Table 2 gels-08-00281-t002:** R^2^ and coefficient values of the zero order, first order, Higuchi model, Korsmeyer–Peppas model and Hixson–Crowell model of PEG/Acrylic Acid hydrogels.

Formulation	Zero Order		First Order		Higuchi Model		Korsmeyer–Peppas Model			Hixson–Crowell Model	
	R^2^	K_0_	R^2^	K_1_	R^2^	KH	R^2^	KKP	n	R^2^	KHC
F1	0.0960	6.696	0.9450	0.273	0.7377	27.448	0.7680	33.430	0.412	0.9231	0.064
F2	0.7782	5.103	0.8561	0.098	0.7413	18.272	0.8346	9.972	0.758	0.8830	0.029
F3	0.8194	5.102	0.9073	0.098	0.8022	18.383	0.8911	10.475	0.740	0.9330	0.029
F4	0.1211	6.697	0.9461	0.274	0.7370	27.475	0.7703	33.705	0.409	0.9224	0.064
F5	0.6195	5.984	0.9540	0.152	0.8829	22.760	0.8992	18.845	0.582	0.9751	0.044
F6	0.8571	5.051	0.9612	0.098	0.8664	18.316	0.9483	10.966	0.719	0.9818	0.029
F7	0.8441	5.627	0.9150	0.114	0.8321	20.304	0.9208	11.707	0.735	0.9465	0.033
F8	0.8170	5.153	0.9244	0.102	0.8133	18.632	0.8973	10.907	0.729	0.9489	0.030
F9	0.7987	6.335	0.9246	0.146	0.8989	23.400	0.9507	16.079	0.662	0.9482	0.042
F10	0.7403	5.500	0.9672	0.124	0.8854	20.508	0.9247	14.946	0.637	0.9843	0.036
F11	0.4296	6.165	0.9687	0.181	0.8760	24.075	0.8768	23.182	0.517	0.9859	0.052
F12	0.4005	5.895	0.9819	0.168	0.8925	23.075	0.8926	22.734	0.507	0.9925	0.048
F13	0.5888	5.793	0.9563	0.148	0.8636	22.122	0.8774	18.586	0.576	0.9774	0.043
F14	0.4440	5.793	0.9888	0.160	0.9115	22.572	0.9123	21.799	0.515	0.9953	0.046
OF	0.4591	5.646	0.9878	0.153	0.9153	21.962	0.9163	21.070	0.518	0.9855	0.044

**Table 3 gels-08-00281-t003:** Formulation design of PEG/Acrylic Acid hydrogels by Design Expert.

Formulation	PEG 6000 (g)	Acrylic Acid (g)	N, N’-Methylene Bis Acrylamide (g)
F1	0.1875	2.5	0.03
F2	0.1250	3.0	0.03
F3	0.2500	2.0	0.03
F4	0.1250	2.0	0.03
F5	0.2500	2.5	0.04
F6	0.1875	2.5	0.03
F7	0.1875	2.0	0.02
F8	0.1250	2.5	0.02
F9	0.2500	3.0	0.03
F10	0.2500	2.5	0.02
F11	0.1250	2.5	0.04
F12	0.1875	2.0	0.04
F13	0.1875	3.0	0,04
F14	0.1875	3.0	0.02

## Data Availability

Not applicable.

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
