# Peer review of "Synthesis of Chemically Cross-Linked pH-Sensitive Hydrogels for the Sustained Delivery of Ezetimibe"

_gels, 2022, doi:10.3390/gels8050281_

Round 1
Reviewer 1 Report
This manuscript described a hydrogel formulation which could be used for ezetimibe delivery. The authors optimized the formulations using PEG, Acrylic Acid, and N, N’-Methylene bis acrylamide with various ratio and did a full characterization. This manuscript is a good fundamental study to understand the specific PEG/Acrylic acid hydrogel physicochemical property for further application. Please address the following comments.
Major comments:
- In swelling study, the author concluded the pH impaction to hydrogel swelling degree. Since the author studied 14 types of hydrogels which with different polymer composition ratio. Is that possible to make a conclusion about polymer ratio impaction to hydrogel swelling degree?
- For Figure 9, the SEM image, which is hard to see the porous structure. Please use “arrow” to indicate the pore structure or use a higher resolution image.
Minor comments:
- In line 76, please indicate the sample name of figure 1A and B in the figure caption.
- In line 97, “Swelling behavior of black hydrogel at various pH.” Please correct “black” to “blank”.
- Please check the grammar of the sentence in Line 139-141.
Author Response
Reviewer 1
This manuscript described a hydrogel formulation which could be used for ezetimibe delivery. The authors optimized the formulations using PEG, Acrylic Acid, and N, N’-Methylene bis acrylamide with various ratio and did a full characterization. This manuscript is a good fundamental study to understand the specific PEG/Acrylic acid hydrogel physicochemical property for further application. Please address the following comments.
Query 1: In swelling study, the author concluded the pH impaction to hydrogel swelling degree. Since the author studied 14 types of hydrogels which with different polymer composition ratio. Is that possible to make a conclusion about polymer ratio impaction to hydrogel swelling degree?
Query reply 1: Respected reviewer, Thanks for your query, Thanks for your query, Yes, it is possible to conclude as with the increase in polymer concentration, swelling properties increase and this was checked by formulating optimized formulation using design expert. Increasing polymer concentration ionized carboxylate group available for ionic repulsive forces which are the major phenomenon behind swelling properties of hydrogels. The supporting references was also mentioned in the revised version of manuscript.
Query 2: For Figure 9, the SEM image, which is hard to see the porous structure. Please use “arrow” to indicate the pore structure or use a higher resolution image.
Query reply 2: Respected reviewer, Thanks for your query, the porous part was highlighted and zoomed images were also added in revised version of file.
Query 3: In line 76, please indicate the sample name of figure 1A and B in the figure caption.
Query reply 3: Respected reviewer, Thanks for your query, Labelling of figure 1 was incorporated in figure caption and intext too in revised version of file.
Query 4: In line 97, “Swelling behavior of black hydrogel at various pH.” Please correct “black” to “blank”.
Query reply 4: Respected reviewer, Thanks for your query, the word was corrected in figure caption.
Query 5: Please check the grammar of the sentence in Line 139-141.
Query reply 5: Respected reviewer, Thanks for your query, all figures were now mentioned intext in revised version of manuscript
Reviewer 2 Report
-The authors should add swelling results at pH 7.4.
-Why the authors have used pH 7.2 for in vitro drug release. It is suggested that the authors need to do in vitro drug release at different pH conditions.
-It should be better to add XRD pattern of the pure hydrogel.
Author Response
Reviewer 2
Query 1: The authors should add swelling results at pH 7.4.
Query reply 1: Respected reviewer, Thanks for your query, we respect your suggestion but we performed swelling studies according to the method mentioned in literature by “Chao, Guo Tao, et al. (DOI: 10.1002/jbm.a.31362).
Query 2: Why the authors have used pH 7.2 for in vitro drug release. It is suggested that the authors need to do in vitro drug release at different pH conditions.
Query reply 2: Respected reviewer, Thanks for your query, drug release was dependent on hydrogels’ degree of swelling and in current study hydrogels showed maximum swelling at pH 7.2, that’s why we performed in-vitro drug release studies at this pH.
Query 3: It should be better to add XRD pattern of the pure hydrogel.
Query reply 3: Respected reviewer, Thanks for your query, XRD analysis was only conducted on pure drug and drug loaded hydrogel. But in future studies we will keep this suggestion in mind and perform XRD on blank formulation as well
Reviewer 3 Report
General comments: The grammar in the manuscript needs to be revised. This is particularly apparent in the introduction section.
Figure 1 – Explain in the caption what A and B correspond to.
Line 87 – “the results” written twice.
Table 1 – Please outline the standard deviations in the obtained results.
Line 99 – KPS mentioned with no prior explanation of abbreviation. The abbreviation is defined on line 249; please switch to define it on first mention.
Line 112 – Full stop should appear after “(Figure 3)”. Likewise with line 126 “(Figure 4)” and subsequent figures.
Figure 3 – Ideally a table earlier in the document explaining the compositions of all tested formulations should be shown.
Lines 133-134 – The comments made here are difficult to interpret given the data shown on Figure 5. This is predominantly due to the formulation coding being non-intuitive. I recommend adding some order to your formulation naming e.g. have the formulations listed in terms of increasing PEG 6000 concentration, then increasing acrylic acid concentration, then increasing crosslinker concentration. So F1 would be 0.1250, 2.0, 0.03 (your current F4) and F2 would be 0.1250, 2.5, 0.04 (your current F11) and so on.
The manuscript found only issues with increasing crosslinker concentration (decreased porosity, decreased entrapment). There should however be some mention of the minimum amount of crosslinker required for the formulation to work. Otherwise, what is the point of having a crosslinker in the system?
Figure 6 – Just like Figure 5, the data here is very difficult to draw conclusions from or understand without more logical ordering of the 15 tested formulations. Please revise the order in which they are being presented throughout the manuscript.
Why was drug release only tested at pH 7.2 and not at pH 1.2 and 6.8? Is the formulation intended for administration via a gastro-resistant tablet/capsule? The fact that swelling studies, etc. were performed at lower pH leads to the logical understanding that drug release at pH 1.2 would also be useful to characterise i.e. to see how much drug is expected to leave the formulation in the 0.5-2 h of gastric transit before it reaches the small intestine.
2.8. – Could you further discuss what best mapping to Hixson-Crowell means for your formulation? This is typically the model observed by dissolving powders, sometimes tablets, and is based on the change in surface area over time. In many cases your data looked to better match first order release, which makes more sense given the formulation in question.
Figure 8 – It would be best to just show the FTIR of (1) unloaded hydrogel, (2) ezetimibe, and (3) the ezetimibe loaded hydrogel. This would allow quick deduction of whether any interactions were present between the drug and the carrier.
Figure 9 – It is difficult to tell what the SEM image is showing. Do the bumps correspond to pores or is it undissolved drug residue? How did the authors confirm this? Are there more SEM images to see what the formulation looks like without drug? Which formulation is the figure actually showing?
What is “OF” in the figures? What is its composition?
Lines 319-326 – Is this a validated method for sol-gel measurement? It feels like after 48 hours in distilled water, even some of the “gel” phase would have dissolved.
Line 379 – Were the samples sputter coated before SEM?
Author Response
Reviewer 3
Query 1: The grammar in the manuscript needs to be revised. This is particularly apparent in the introduction section.
Query reply 1: Respected reviewer, Thanks for your query, thanks for the query we rechecked the manuscript from English native speaker to remove grammatical errors.
Query 2: Figure 1 – Explain in the caption what A and B correspond to.
Query reply 2: Respected reviewer, Thanks for your query, figure captions added in revised file.
Query 3: Line 87 – “the results” written twice.
Query reply 3: Respected reviewer, Thanks for your query, duplication removed in the said line.
Query 4: Table 1 – Please outline the standard deviations in the obtained results.
Query reply 4: Respected reviewer, Thanks for your query, in table 1 we mentioned predicted results obtained from design expert and obtained results of optimized hydrogel. As there is only one optimized hydrogel that’s why we did not mentioned any SD values.
Query 5: Line 99 – KPS mentioned with no prior explanation of abbreviation. The abbreviation is defined on line 249; please switch to define it on first mention.
Query reply 5: Respected reviewer, Thanks for your query, as per your kind suggestion we updated the abbreviation in said lines for KPS.
Query 6: Line 112 – Full stop should appear after “(Figure 3)”. Likewise with line 126 “(Figure 4)” and subsequent figures.
Query reply 6: Respected reviewer, Thanks for your query, we modified the issues in whole manuscript file.
Query 7: Figure 3 – Ideally a table earlier in the document explaining the compositions of all tested formulations should be shown.
Query reply 7: Respected reviewer, Thanks for your query, as per journal guidelines methodology section must be mentioned after results section that’s why tables explaining the composition of prepared hydrogel formulations were placed later.
Query 8: Lines 133-134 – The comments made here are difficult to interpret given the data shown on Figure 5. This is predominantly due to the formulation coding being non-intuitive. I recommend adding some order to your formulation naming e.g. have the formulations listed in terms of increasing PEG 6000 concentration, then increasing acrylic acid concentration, then increasing crosslinker concentration. So F1 would be 0.1250, 2.0, 0.03 (your current F4) and F2 would be 0.1250, 2.5, 0.04 (your current F11) and so on.
Query reply 8: Respected reviewer, Thanks for your query, we agreed with your suggestion but formulation concentration was obtained using design expert software and formulations names were also attributed by software with respect to variables such as polymer, crosslinker and monomer. That’s why we used the formulation naming automatically indicated by design expert.
Query 9: The manuscript found only issues with increasing crosslinker concentration (decreased porosity, decreased entrapment). There should however be some mention of the minimum amount of crosslinker required for the formulation to work. Otherwise, what is the point of having a crosslinker in the system?
Query reply 9: Respected reviewer, Thanks for your query, we perform preformulation studies to set concentrations of polymer, crosslinker and monomer and from those results we set the concentration ranges in design expert and software suggested different fourteen formulations. The concentration of crosslinker depends on the amount of polymer used and minimum concentration required for cross-linker was 0.02 g and a maximum of 0.04g.
Query 10: Figure 6 – Just like Figure 5, the data here is very difficult to draw conclusions from or understand without more logical ordering of the 15 tested formulations. Please revise the order in which they are being presented throughout the manuscript.
Query reply 10: Respected reviewer, Thanks for your query, we agreed with your query but as we mentioned earlier in your query, the formulation naming was automatically indicated by design expert and we use same in the whole manuscript.
Query 11: Why was drug release only tested at pH 7.2 and not at pH 1.2 and 6.8? Is the formulation intended for administration via a gastro-resistant tablet/capsule? The fact that swelling studies, etc. were performed at lower pH leads to the logical understanding that drug release at pH 1.2 would also be useful to characterise i.e. to see how much drug is expected to leave the formulation in the 0.5-2 h of gastric transit before it reaches the small intestine.
Query reply 11: Respected reviewer, Thanks for your query, No the formulation was not intended for administration via a gastro-resistant tablet/capsule. Swelling studies were conducted to check at what pH formulation showed maximum swelling. As targeted intestinal pH was 7.2 and formulation also showed maximum swelling at this pH, that’s why we only performed drug release studies according to the study objective. Lastly, the results of swelling studies suggested that hydrogels do not swell at pH 1.2, that’s why it is capable of bypassing gastric mucosa and releasing its contents at intestinal pH.
Query 12: 2.8. – Could you further discuss what best mapping to Hixson-Crowell means for your formulation? This is typically the model observed by dissolving powders, sometimes tablets, and is based on the change in surface area over time. In many cases your data looked to better match first order release, which makes more sense given the formulation in question.
Query reply 12: Respected reviewer, Thanks for your query, the Hixson-Crowell model indicate changes in surface area with respect to time. The hydrogel formulations swelled as dissolution time increased and showed an increased in surface area respectively. That’s why the release pattern corresponds to Hixson-Crowell model. The first order kinetics revealed that the drug release from formulation was dependent on initial concentration.
Query 13: Figure 8 – It would be best to just show the FTIR of (1) unloaded hydrogel, (2) ezetimibe, and (3) the ezetimibe loaded hydrogel. This would allow quick deduction of whether any interactions were present between the drug and the carrier.
Query reply 13: Respected reviewer, Thanks for your query, as per your kind suggestion we modified the FTIR spectra figure.
Query 14: Figure 9 – It is difficult to tell what the SEM image is showing. Do the bumps correspond to pores or is it undissolved drug residue? How did the authors confirm this? Are there more SEM images to see what the formulation looks like without drug? Which formulation is the figure actually showing?
Query reply 14: Respected reviewer, Thanks for your query, the SEM image was from optimized formulation and image showed that hydrogel has a porous structure with coarse surface. The SEM image was labelled and zoomed images were also shown in revised version of file.
Query 15: What is “OF” in the figures? What is its composition?
Query reply 15: Respected reviewer, Thanks for your query, OF is optimized formulation and its composition was mentioned in method section under numerical optimization heading performed using design expert.
Query 16: Lines 319-326 – Is this a validated method for sol-gel measurement? It feels like after 48 hours in distilled water, even some of the “gel” phase would have dissolved.
Query reply 16: Respected reviewer, Thanks for your query, Yes, this method was previously used by Zhao, Long, et al. (doi.org/10.1016/S0144-8617(03)00103-6) and reference was also added in method section.
Query 17: Line 379 – Were the samples sputter coated before SEM?
Query reply 17: Respected reviewer, Thanks for your query, no samples were not sputter coated but lyophilized before SEM.
Round 2
Reviewer 2 Report
The manuscript is acceptable in its current form.